# Targeted Proteomics for Monitoring One-Carbon Metabolism in Liver Diseases

**DOI:** 10.3390/metabo12090779

**Published:** 2022-08-24

**Authors:** Laura Guerrero, Alberto Paradela, Fernando J. Corrales

**Affiliations:** 1Centro Nacional de Biotecnología (CNB), CSIC. C/Darwin 3, 28049 Madrid, Spain; 2National Institute for the Study of Liver and Gastrointestinal Diseases (CIBERehd, Carlos III Health Institute), 28029 Madrid, Spain

**Keywords:** liver disease, hepatocellular carcinoma, one-carbon metabolism, proteomics

## Abstract

Liver diseases cause approximately 2 million deaths per year worldwide and had an increasing incidence during the last decade. Risk factors for liver diseases include alcohol consumption, obesity, diabetes, the intake of hepatotoxic substances like aflatoxin, viral infection, and genetic determinants. Liver cancer is the sixth most prevalent cancer and the third in mortality (second in males). The low survival rate (less than 20% in 5 years) is partially explained by the late diagnosis, which remarks the need for new early molecular biomarkers. One-carbon metabolism integrates folate and methionine cycles and participates in essential cell processes such as redox homeostasis maintenance and the regulation of methylation reactions through the production of intermediate metabolites such as cysteine and S-Adenosylmethionine. One-carbon metabolism has a tissue specific configuration, and in the liver, the participating enzymes are abundantly expressed—a requirement to maintain hepatocyte differentiation. Targeted proteomics studies have revealed significant differences in hepatocellular carcinoma and cirrhosis, suggesting that monitoring one-carbon metabolism enzymes can be useful for stratification of liver disease patients and to develop precision medicine strategies for their clinical management. Here, reprogramming of one-carbon metabolism in liver diseases is described and the role of mass spectrometry to follow-up these alterations is discussed.

## 1. Liver Diseases

The increasing incidence of liver diseases represents a heavy burden on the global population and health systems, accounting for more than 2 million deaths per year worldwide [1]. Some risk factors for liver diseases are well known, such as alcohol consumption, infection by hepatitis virus A, B, and C (mainly), obesity, diabetes, the intake of hepatotoxins like aflatoxin, and genetic determinants, among others [2,3,4,5].

The increasing incidence of liver diseases is in part due to the adoption of a more sedentary lifestyle and to habits including alcohol abuse or a high fat diet. More than 2 billion people worldwide are alcohol consumers (43% of global population aged more than 15 years) [6] of which 75 million have been diagnosed with alcohol associated disorders. Regions with higher rates of alcohol consumers over 15 years old among the total population are Europe (59.9%), America (54.1%), and Western Pacific region (53.8%).

In addition to alcohol consumption, obesity is a major risk factor for liver damage. Obesity can be associated to metabolic alterations resulting from specific genetic determinants or to dietary factors including hypercaloric diets with high fat and carbohydrate content. The incidence of obesity has increased 6-fold over the last 40 years [7] and often leads to metabolic syndrome [8,9], non-alcoholic fatty liver disease (NAFLD), and non-alcoholic steatohepatitis (NASH) [10]. About 2 billion people are obese worldwide, and more than 400 million are diabetic, both pathologies being risk factors for NAFLD. NAFLD is the most common liver disease in Western countries [10] with a global prevalence of 25.24% [4], and it is a recognized risk factor for liver cancer development [11,12].

Viral hepatitis resulting from the infection of hepatitis A, B, C, D and E viruses (HAV, HBV, HCV, HDV, and HEV) are also prevalent liver diseases [13]. Viral hepatitis frequently progresses to cirrhosis and HCC. It caused 1.34 million deaths in 2015, ranking as the 7th most common cause of death [14] mainly in Asia and sub-Saharan Africa. Effective strategies against viral hepatitis include vaccination for HAV and HBV and antiviral treatments for HCV like boceprevir, telaprevir, simeprevir, and sofosbuvir [15,16]. However, despite the availability of these preventive and therapeutic options, viral hepatitis represents a major concern in developing countries.

Liver homeostasis can be also impaired by genetic determinants as is the case of several disease conditions generically termed as progressive familiar intrahepatic cholestasis (PFIC) [17]. Cholestasis is the accumulation of bile acids outside the gallbladder, and it can be intrahepatic or extrahepatic depending on where the bile acid accumulation occurs. Under physiological conditions, bile acids are synthetized in the liver, stored at the gallbladder, and then released to the small intestine (duodenum) where they play a key role in the regulation of the absorption of liposoluble vitamins, cholesterol, and lipids. Cholestasis can be caused by obstructions of bile flow produced by gallstones, primary sclerosing cholangitis, tumours in bile ducts, or pregnancy, among others. Hereditary cholestasis involves a group of rare autosomal recessive liver disorders caused by alterations in genes related to the secretion and transports of bile [18]. PFIC consists of an intrahepatic accumulation of free bile acids that produce severe and chronic liver damage since early ages and make liver transplantation necessary in most cases [18,19]. Several mutated genes have been associated to PFIC, including transporters *atp8b1* [20,21,22], *abcb11* [23,24], *abcb4/mdr3* [25,26,27,28] or *tjp2* [18].

Liver cancer is the sixth most frequent type of cancer, the third in terms of mortality (the second for males), and it has an increasing incidence over the last decades [29]. The most frequent subtype is hepatocellular carcinoma (HCC), representing 90% of liver cancer cases, followed by cholangiocarcinoma. HCC main risk factors include alcohol consumption, viral hepatitis, steatohepatitis, NAFLD, obesity, hepatotoxins intake like aflatoxin, and genetic determinants [12,30] (Figure 1). HCC incidence is heterogeneous: most cases occur in Asia (72%) (mainly in China), followed by Europe (19%), Africa (7.8%), North America (4.6%), South America (4.6%), and Oceania (0.5%) [30]. Mediterranean countries have intermediate incidence of HCC: between 10 to 20 cases per 100,000 individuals, while America have lower rates of HCC (<5/100,000) [31]. HCC etiology varies according to the prevalence of specific risk factors on each geographical region. Sub-Saharan Africa and Eastern Asia are endemic regions for HBV infection and the incidence rate there is over 20 cases per 100,000 individuals. From the above-mentioned risk factors, HBV, HCV, and alcohol consumption account for about 90% of the cases as follows: 40% by HBV, 40% by HCV, and 10% by abusive alcohol consumption. In all cases, chronic liver injury leads to a progressive loss of function and to the onset of a sequential liver damage that involves inflammation, steatosis, fibrosis, and cirrhosis. The last one is considered as a pre-tumoral condition since above 80% of HCC cases develop on cirrhotic livers [31]. This represents an important hurdle in the clinical management of HCC compared to other malignancies since liver function is severely impaired, and the treatment options are limited to those most conservative with the damaged liver parenchyma. The clinical strategies to treat HCC depend on the stage of the disease, but frequently include resection of the tumour if it is localized and liver function is preserved, liver transplantation, transarterial chemoembolization (TACE), or chemotherapeutic drugs like atezolizumab, bevacizumab, or sorafenib [32,33,34,35]. Despite the active investigation to elucidate the molecular basis of HCC pathogenesis and the development of innovative treatment options, HCC prognosis is poor, with a mean survival of 6–20 months [36] and a 5-years survival rate remaining around 10% of cases [37]. The reasons for this poor outcome are varied, but late diagnosis is a main factor as evidenced by the observation that five-year survival can exceed 70% of cases when diagnosis is made at early stages [38]. This highlights the urgent need for biomarkers for early tumour detection as well as to foresee the outcome and the response to drugs, integrating a precise and personalized strategy to fight HCC.

## 2. The Role of Proteomics in Biomarker Discovery and Validation

Biomarkers are parameters or molecules of a different chemical nature (nucleic acids, lipids, proteins, or other metabolites) that would provide objective measurements of a specific condition (e.g., disease) and can be eventually used in the clinic to stratify patients, ideally providing high specificity and sensitivity. According to the purpose of use, a biomarker should meet certain requirements. For diagnosis, the biomarker should be detected in asymptomatic patients at early stages of a disease, preventing progression to more severe stages, when the therapeutic options might be limited. A prognosis biomarker should predict the progression of a disease to implement the most suitable follow-up strategy.

Biomarkers can be genetic mutations like single nucleotide polymorphisms (SPNs), a well-known example of this is the association between *brca1* gene polymorphism and breast cancer risk [39]. In addition to mutations on specific genes, microRNAs can be used as biomarkers, an example of this is the downregulation of miR-122, miR-34a, mi-R-16, and miR-21 in patients with liver fibrosis in a stage-dependent way [40,41]. Finally, there are biomarkers to predict and evaluate the response of a patient to a specific therapy, and therefore, to allow interventions on a personalized medicine basis [42].

Biomarkers are especially relevant for early diagnosis of diseases with a high prevalence and mortality, such as some of the above-mentioned liver diseases. There are recognized biomarkers to assess liver function and disease in the clinical routine, such as serum aspartate aminotransferase (AST) and alanine aminotransferase (ALT) [43]. AST and ALT are produced mainly in the liver and the increased presence of these enzymes in the serum directly correlates with liver damage progression. Moreover, combinations of serum proteins and metabolites can be used to determine the severity of certain liver alterations as, for instance, FibroTest, that assess liver fibrosis stage using an algorithm that integrates the quantification of five serum biomarkers easily measured in any clinical biochemistry laboratory: alpha-2-macroglobulin, apolipoprotein A1, haptoglobin, γ-glutamyltranspeptidase, and bilirubin [44]. In the same direction, other non-invasive method, FibroScan, has been developed in the last decades for evaluating liver stiffness. This procedure uses an ultrasound-based technique known as transient elastography, that measures the speed of propagation of elastic waves through the liver, which is directly correlated to liver stiffness and fibrosis [45]. The combination of FibroTest and FibroScan can be used to accurately establish the degree of liver fibrosis in patients with HCV [46]. Despite this positive progress, the development of new specific molecular biomarkers for the clinical management of liver diseases that may boost the development of personalized medicine strategies remains a major need.

On the postgenomic era, proteomics plays a fundamental role on biomarker discovery [47,48]. The technologic progress of mass spectrometry during the last years has allowed the simultaneous identification and quantification of thousands of proteins in complex biological samples. New mass spectrometers include high resolution and increased sensitivity, allowing the identification and quantification of hundreds of proteins from very limited amounts of sample [49]. This has made possible single-cell proteome analyses [50], which has the potential to dissect the tissue heterogeneity at the cellular level. This is especially relevant in cancer, where intra-tumoral heterogeneity might compromises efficient responses to a given therapy [51].

Mass spectrometry techniques can be applied for biomarker discovery and validation. Common proteomics workflows for biomarker discovery typically integrates two different stages (Figure 2). The first stage relies on shotgun proteomics, where thousands of proteins are identified and quantified in a single experiment [52,53]. Proteins are digested to peptides and the resulting mixture is analyzed by high-performance liquid chromatography (HPLC) coupled to mass spectrometry. According to the availability of sample and the analytical requirements, different chromatography setups can be used (nano, micro, etc.). Data acquisition can be made in two different modes: data dependent acquisition (DDA), where precursor ions (i.e., peptides) that comply with predefined conditions (e.g., *m*/*z* within a given range) and have a high signal in a first full scan are selected for fragmentation generating tandem mass spectra (MS/MS) that are posteriorly matched against a sequence data base for peptide identification. The main disadvantage of this type of acquisition is that peptides from low abundant proteins have low intensity signals and sometimes are not selected for fragmentation; thus, some information can be lost. Alternatively, data independent acquisition (DIA) allows for recording virtually all signals as every precursor ion is selected for fragmentation [54]. DIA generates vast amount of data, and these experiments are usually analyzed using spectral library databases, increasing the computational cost. To increase the proteome coverage, several methods can be used, including sample prefractionation or ion mobility modules that improve the separation of peptides according to their physicochemical features. However, while allowing for efficient molecular screening, shotgun proteomics is based on complex protocols that limit the analytical sample throughput, mainly if ample proteome coverages are needed. Therefore, complementary protocols are necessary to validate the candidate biomarkers and to develop and standardize measurement methods compatible with the clinical routine. In this context, targeted proteomics has emerged as a reference method for biomarker validation, as it provides for the accurate quantification of a predefined list of proteins [55,56]. There are two main approaches to target and quantify selected proteins by mass spectrometry: multiple reaction monitoring (MRM) and Parallel Reaction Monitoring (PRM). The first one is performed in triple quadrupole instruments (QqQ, low resolution) and allows for quantification of selected protein-specific peptides by sequentially monitoring the precursor ion and a few fragments (called transitions) produced in the collision cell during the mass spectrometry analysis. The quantification of several independent peptides and transitions provides an accurate estimation of the abundance of the targeted protein. PRM is performed in high resolution instruments and a signal of all fragments for each selected precursor (peptide) is obtained. Targeted proteomics specifically monitors proteins of interest similarly as other techniques do (e.g., affinity-based strategies), but it circumvents the well-known limitations of antibodies, which are expensive, not always available, and frequently cross-react nonspecifically with unwanted ligands. Another main advantage of targeted proteomics is its multiplexing capacity, allowing the simultaneous quantification of tens of proteins across multiple samples. This is especially useful for high throughput biomarker validation on large cohorts.

Proteomics has been extensively used for the study of liver physiology and pathology, and these studies have led to the proposal of a wide panel of protein biomarkers for the diagnostics and follow-up of different liver diseases, including hepatocellular carcinoma [57]. Proteomic analysis of the serum of HCC patients has allowed the identification of HCC hallmarks like a C-terminal fragment of complement C3 protein and isoform of APOA1 [58]. Furthermore, serum levels of cytokeratin 19 fragment CYFRA21-1 have been associated with metastasis and aggressiveness of HCC [59]. The liver is an organ with high metabolic activity, and alterations of the hepatocyte metabolism parallels the progression of the chronic liver disease up to the onset of HCC. Upon tumoral transformation, the energy production in hepatocytes switches from mitochondrial oxidative phosphorylation to anaerobic glycolysis, a phenomenon known as Warburg effect [60]. This metabolic reprograming helps the tumoral cell to use a more versatile source of nutrients to produce energy and to promote cell growth and proliferation [61]. Beyond the Warburg effect, metabolic reprogramming affects to additional interconnected pathways such as one-carbon metabolism (OCM) cycle, essential to maintain hepatocytes in a quiescent and differentiated state. OCM has a specific configuration in the liver and must be regulated in oncogenesis to cope with the tumoral cell requirements of one-carbon units for nucleotide synthesis for cell division and for methylation reactions, among others [62].

## 3. One-Carbon Metabolism

One-carbon metabolism (Figure 3) is a metabolic pathway that recycles one carbon units across methionine and folate cycles. This activity is essential for the processing of amino acids, glucose, and vitamins as well as for the biosynthesis of macromolecules (lipids, nucleotides, and proteins), for the maintenance of redox homeostasis and for the balance of the methylation reactions [63,64], which makes OCM the bridge between intermediate metabolism and epigenetic regulation [65]. OCM reactions occur in several subcellular compartments [66], and some of the participant enzymes are distributed across different cellular organelles. For instance, serine hydroxymethyltransferase (SHMT) isoforms 1 and 2 catalyze the transference reaction of the beta carbon from serine to THF to produce glycine and 5,10 methylene THF in the cytosol and in the mitochondria respectively [67].

Diverse biomolecules such as vitamins (folate), amino acids (methionine or serine), or intermediate products from glucose metabolism like 3-Phosphoglyceric acid (3PG) feed the one-carbon metabolism cycle [68]. On the other side, the by-products of OCM are essential for a wide variety of cellular processes. As an example, tetrahydrofolate (THF) is synthetized from folate in a reaction mediated by the dihydrofolate reductase enzyme (DHFR). THF is then used for the formation of 5,10 methylenetetrahydrofolate (5,10 CH_2_-THF) that is subsequently transformed in 10 formyl-tetrahydrofolate by the dehydrogenation mediated by methylenetetrahydrofolate dehydrogenase (MTHFD). Interestingly, 10 formyl-tetrahydrofolate is a precursor for purine synthesis, which is essential [69] for cell growth and proliferation, and in fact, it has been largely recognized as a target for cancer therapy. Accordingly, antifolates are one of the first anticancer drugs described and were synthetized as early as in 1947, proving its effect in remission of child leukaemia [69]. 5,10 CH_2_-THF can also act as substrate of the methylenetetrahydrofolate reductase to produce methyltetrahydrofolate (5-CH_3_–THF) that serves as methyl group donor to recycle back homocysteine to methionine in a reaction catalyzed by methionine synthase (MS).

Methionine and ATP are used to produce S-adenosylmethionine (SAM) in a reaction catalyzed by methionine adenosyl transferases (MAT). There are different MAT isoforms, which are products of different genes, being MAT1A specific of the adult liver while MAT2A is ubiquitously expressed in other tissues as well as in fetal liver [70]. In the next step of the methionine cycle, glycine N-methyl transferase catalyzes the transference of the methyl group of SAM to glycine to produce sarcosine and S-adenosylhomocysteine (SAH), that can be further metabolized to homocysteine by the adenosylhomocysteinase. Homocysteine is involved in the maintenance of the redox balance of the cell through the trans-sulphuration pathway, leading to the production of glutathione. First, cystathionine β synthase (CBS) catalyzes the production of cystathionine from homocysteine, which is then metabolized into cysteine that can be used for protein synthesis or to produce glutathione, which is the main reducing agent in hepatocytes [71]. In humans, the deficiency of CBS induces impaired cognitive development, Marfan’s syndrome-like connective tissue, vascular and renal defects, thrombosis, and abnormalities like hepatic fat accumulation, and increased size of the liver [72]. Moreover, accumulation of homocysteine is a recognized risk factor for cardiovascular diseases [73] as well as for spina bifida in newborns [74].

Methionine cycle is also involved in the biosynthesis of phospholipids [75]. Phosphatidyl choline, one of the main components of the cellular membrane [76], is synthetized through three sequential methylations of phosphatidylethanolamine where SAM donates the methyl groups [77]. SAM is the main donor of methyl groups in methylation reactions of the cell like histone or nucleic acids methylation, thus regulating epigenetics [78,79]. Upon decarboxylation, the propylamine group of SAM can be used for polyamine synthesis leading to the synthesis of spermidine and spermine in two consecutive reactions where methylthioadenosine (MTA) is also produced. This metabolite enters the methionine salvage pathway in a reaction catalyzed by methylthioadenosine phosphorylase (MTAP) and it is recycled back into methionine.

OCM has been extensively studied in the liver, where it has a tissue specific architecture. The metabolic balance of OCM must be precisely regulated in the liver, as it has been shown that tiny alterations lead to loss of function and dedifferentiation of hepatocytes. In mammals, there are two genes encoding MAT enzymes, MAT1A and MAT2A [80]. MAT1A is mostly expressed in differentiated hepatocytes whereas MAT2A is expressed in the fetal liver, in extrahepatic tissues, and in liver cancer. In contrast to MAT2A, MAT1A has a high methionine metabolic capacity resulting from its activation by this amino acid and the lack of inhibition by the product SAM [81]. Moreover, MAT1A has a specific regulatory mechanism based on the redox regulation of its activity (that prevents excessive ATP consumption and cellular de-energization in critical liver injury processes [82]. Preservation of high MAT1A and low MAT2A expression levels is essential to maintain a functional and differentiated liver [83]. Experimental evidence demonstrates that liver MAT1A expression is associated with hypomethylation and elevated levels of chromatin acetylation, while its silencing leads to hepatocyte proliferation and liver tumor progression [84]. In turn, SAM has a key role in preserving MAT1A expression and preventing the switch to MAT2A, but its accumulation also leads to HCC development [85]. For this reason, it appears that a precise regulation of the SAM intracellular content is essential for the preservation of hepatocyte homeostasis. In line with this idea, it has been demonstrated that SAM has a protective role against liver insults compromising its synthesis [86,87] and, more interestingly, its administration improves the condition of Child 2 cirrhotic patients [88,89,90].

Other intermediate metabolites participating in the one-carbon metabolism pathway have also been studied as potential therapeutic agents for liver diseases. 5-methylthioadenosine (MTA) is involved in the regulation of gene expression, inflammation, proliferation, differentiation, and apoptosis and its levels must be precisely regulated. In rat model of chronic liver damage induced by CCl_4_, MTA has also proved to have an hepatoprotective role showing a strong antioxidant effect and reducing liver cell damage and fibrosis [91]. The hepatoprotective role of MTA has also been proved to combat bile acids injury in cholestasis models. *Mdr2/abcb4* deficient mice lack the canalicular phosphatidylcholine flippase [92] and is a remarkable murine model of PFIC3. Phosphatidyl choline is essential for the formation of the micelles where bile acids are stored, and the lack of phosphatidylcholine at the bile canaliculus causes the accumulation of free bile acids that produce severe tissue damage since early ages. Inflammation in *Mdr2* −/− mice occurs since week 2–3, periportal fibrosis occurs at week 4, and preneoplastic lesions appear at month 4–6. In this model of liver injury, MTA reduces the production of proinflammatory and pro-fibrotic mediators and inhibits the proliferation and activation of fibrogenic cells, proving its protective role from liver injury [93]. This suggests that MTA could be useful for the treatment of PFIC and other cholestatic disorders that produce liver fibrosis.

## 4. One-Carbon Metabolism in NAFLD, NASH and Fibrosis

Non-alcoholic fatty liver disease (NAFLD) consists of an accumulation of triglycerides in the liver (hepatic steatosis) not associated with alcohol consumption, and often correlated with obesity and metabolic syndrome [94]. NAFLD affects to more than 25% of the population worldwide and it has an increasing incidence due to hypercaloric diets and sedentarism [10,95].

NAFLD can progress to non-alcoholic steatohepatitis (NASH), a more severe condition where inflammation plays a central role in the liver response to injury and contributes to its chronicity. In most cases, hepatic steatosis is reversible and does not have serious complications, but individuals with NASH are at higher risk of developing fibrosis, and, eventually, cirrhosis and HCC [94]. In fact, cirrhosis develops in 25% of patients with NAFLD [96].

Alterations in one-carbon metabolism are associated with NAFLD [97]. Phosphatidylcholine (PC) is one of the most abundant phospholipids in the cell, and it participates in cell membranes structure, bile acids micelles formation, and fats transport; processes that are impaired under a PC deficiency leading to triglyceride accumulation About 30% of PC in the hepatocyte is synthetized through the transference of the methyl group of SAM to the phosphatidylethanolamine [98] (Figure 4). This methylation reaction is catalyzed by phosphatidylethanolamine methyl transferase (PEMT), and it is the link between the OCM and lipid metabolism in the liver [99]. SAM is produced from methionine in a reaction catalyzed by MAT. SAM levels in the liver must be precisely regulated to avoid liver damage [100]. Accordingly, a dietary deficiency of methionine, precursor of SAM, or choline, is known to induce hepatic steatosis since 1964 [101]. Indeed, impairment of SAM synthesis leads to lipid accumulation and NASH as demonstrated in the MAT1A −/− mouse model [102]. Similarly, impairment of PC synthesis in PEMT deficient mice increased the sensitivity to develop NAFLD in response to high fat diet [103]. Besides, high fat diets, which inhibit SAM synthesis, induce a depletion of PC and a concomitant reduction of lipoprotein secretion, thus increasing the accumulation of triglycerides in the liver [104,105]. Therefore, the maintenance of the intracellular levels of SAM appears as a central issue to preserve a normal liver function and to prevent disease progression.

In addition to these factors directly involved in PC synthesis, alterations of other OCM enzymes can be also associated with NAFLD and NASH. Disfunction of CBS causes an accumulation of homocysteine and hyperhomocysteinemia, which has been correlated to fatty liver in patients [106] and in high fat diet mouse models [107]. Supporting this correlation, CBS deficiency (CBS −/− mice) alters liver morphology and induce microvesicular steatosis [107]. In addition, recycling homocysteine to methionine mediated by of BHMT could be a preventive option. BHMT catalyzes the formation of methionine from betaine and homocysteine, reducing homocysteine levels. In a rat model of NAFLD induced by high fat diet, betaine supplementation has proved to prevent liver accumulation of triglycerides and liver damage and inflammation through the regulation of BHMT, GNMT and monoacylglycerol acyltransferase (MGAT) [108]. However, despite the evident correlation between accumulation of homocysteine and fat in the liver, the molecular mechanisms involved still remain unclear.

Another OCM enzyme that has been associated with NAFLD is glycine N-methyltransferase (GNMT). This enzyme catalyzes the transference of the methyl group of SAM to glycine to produce sarcosine, thus regulating SAM levels. GNMT is a highly abundant enzyme in the liver, pancreas, and prostate, and is absent in HCC. Individuals with GNMT deficiency or mutated GNMT showed elevated serum levels of aminotransferases, methionine and SAM, and developed steatosis and fibrosis [85,109]. Moreover, GNMT −/− mice developed fatty liver, fibrosis, and HCC [85], demonstrating that the regulation of SAM levels in the liver is important to prevent steatosis and progression to HCC.

An essential measure in the treatment of obese patients with NAFLD is to promote weight loss. This can be achieved by a combination of dietary caloric restrictions and by increasing physical activity [110]. Pharmacologic strategies have been also developed, including the administration of pioglitazone, and they have proved to be safe and effective [111]. Additionally, in light of the association between OCM and NAFLD, additional therapeutic options might emerge from OCM intermediates like SAM or betaine.

## 5. One-Carbon Metabolism in Cirrhosis and Hepatocellular Carcinoma

Cirrhosis and hepatocellular carcinoma (HCC) rank 11th and 16th among the most frequent cause of death, respectively. In combination, they caused 3% of all deaths worldwide in 2000 and 3.5% in 2015, which remarks their increasing incidence. In terms of morbidity, cirrhosis disease causes 1.5% of disability, representing a severe economic impact worldwide [112]. Cirrhosis commonly results from chronic liver diseases, and is one of the main effects of HBV and HCV infection and abusive alcohol consumption. It frequently progresses to HCC, which is the main type of liver cancer in such cases [113]. Early diagnosis of HCC can achieve up to 70% of survival after 5-years upon liver transplant, tumor resection, or chemotherapy [114]. Unfortunately, most cases are diagnosed too late and 5-years survival rate of HCC commonly drops to 12% [115]. It is worth noting that more than 90% of HCC cases progress from a cirrhotic liver [33], which remarks the importance of monitoring cirrhotic patients for an early detection.

Methionine metabolism is significantly impaired in cirrhosis, leading to hypermethioninemia, hyperhomocysteinemia and reduced production of glutathione in the liver. The molecular basis lay on the reduced expression of several OCM enzymes including MAT1A, GNMT, BHMT, CBS, and MS in liver cirrhosis. SAM synthesis drops since the expression of MAT1A is significantly reduced in cirrhosis through a mechanism dependent on the hypermethylation of its promoter [116]. Besides, MS activity is also reduced compromising the remethylation capacity of homocysteine to methionine. While in some cases BHMT compensates for MS deficiency, persistent liver injury also abrogates this alternative reaction for homocysteine remethylation leading to a drop of SAM content in hepatocytes, whereas SAH and homocysteine levels are increased. The limited capacity to produce SAM compromises the methylation of biomolecules in hepatocytes, modify epigenetic landmarks in proteins and DNA, leading to a reprogramming of gene expression and to liver disease progression [117].

One-carbon metabolism has been extensively studied in HCC, and evidence has been accumulated that demonstrate its essential function to maintain fully functional and differentiated hepatocytes. As in cirrhosis, alterations in OCM enzymes or intermediate metabolites contribute to HCC progression [116]. OCM enzymes have a tissue specific expression pattern, being abundantly expressed in the liver. During HCC progression, this liver specific pattern is lost and OCM is significantly reprogrammed to cope with the requirements of actively proliferating cells [118].

SAM is the main donor of methyl groups in methylation reactions of the cell and its levels must be precisely regulated to maintain liver function and differentiation. SAM is produced from methionine through a reaction catalyzed by methionine adenosyltranferases. In mature liver the main isoform of this enzyme is MAT1A, while MAT2A is expressed is other tissues. In HCC, the loss of hepatocyte differentiation is associated with a downregulation of MAT1A expression and an up-regulation of MAT2A known as MAT1A:MAT2A switch [119]. The balance between MAT1A and MAT2A has been related to increased metastatic activity and poor prognosis of HCC through the activation of RAS/ERK (Rat sarcoma/Extracellular regukated kinase), PI3K/AKT (Phosphatidylinositol-3-kinase/Serine-threonine protein kinase) and NF-κB (Nuclear Factor NF-kappa-B) pathways, providing proliferative and survival advantages to tumoral cells [120].

Glycine N-methyltransferase is highly expressed in the liver, and it is not expressed in HCC. Furthermore, GNMT mRNA levels are significantly lower in patients at risk of developing HCC, thus, GNMT has been proposed as a tumor-susceptibility gene for liver cancer. GNMT has high affinity for SAM and plays an essential function in maintaining its intracellular levels under a very precise control. This regulatory mechanism is needed for normal hepatocyte function, in fact, mice lacking GNMT have develop a severe liver disorder with high serum levels of aminotransferase, methionine, and SAM, and ultimately develop HCC [85,121]. The loss of GNMT produces an increase of SAM levels that induces aberrant DNA and histones methylation, causing alterations in epigenetic regulation that results in tumor progression in mice liver [85]. Besides, in GNMT −/− tumors, cell pathways involved in cell proliferation and tumor formation including Ras and JAK/STAT (Janus kinase/Signal Transducer and Activator of Transcription) signaling are activated, and suppressors of these pathways (RASSF1, RASSF4, SOCS and cytokine-inducible SH2-protein) are inhibited.

Methylthioadenosine phosphorylase (MTAP) is an enzyme expressed ubiquitously that participates in the methionine salvage pathway, but the highest expression levels are in the liver [122]. MTAP catalyzes the hydrolysis of methylthioadenosine (MTA) to produce 5-methylthioribose-1-phosphate and adenine, that participates in methionine and purine salvage pathways, respectively. Proteome and methyl-proteome wide studies revealed that MTAP deficiency induces an intracellular accumulation of MTA and might be considered as a bad prognosis since it provides a proliferative advantage and produces resistance to apoptosis and chemotherapeutic drugs [123].

In summary, maintenance of the proper expression levels of OCM enzymes and metabolites is essential for the preservation of the quiescent and differentiated state of the hepatocyte. The loss of function or silencing of OCM enzymes like MAT1A, GNMT, or MTAP leads to the progression of HCC. Therefore, OCM enzymes are a potential source for HCC biomarkers of the liver function and differentiation.

## 6. Proteomics for Biomarker Discovery. Monitoring of OCM in Liver Diseases

Mass spectrometry has been widely used to study the molecular mechanisms underlying the progression of liver disease. Given its high impact and high prevalence, the proteome of liver diseases such as HCC has been extensively studied. Diverse proteomics techniques like label free shotgun proteomics, isobaric labelling, or targeted proteomics have been applied to the identification and quantification of proteins with differential expression or posttranslational modification patterns in HCC. For instance, Jiang et al. have recently identified new therapeutic targets for early-stage HCC. This study involved 110 HCC associated to HBV infection that could be stratified into three subtypes. Massive proteomic and phosphoproteomic analysis revealed that the subtype with disrupted cholesterol homeostasis and high levels of sterol O-acyltransferase (SOAT1) was associated with the lowest rate of survival and the greatest risk of a poor prognosis after surgery. Furthermore, suppression of SOAT1 inhibits the proliferation and migration of HCC cells and the treatment with the SOAT1 inhibitor avasimibe reduced the size of tumors in patient-derived tumour xenograft mouse models [124].

Proteomics has also been applied to seek for plasma biomarkers of liver disease, including NAFLD. Plasma proteome profiling of 48 patients with and without NAFLD and normal glucose tolerance, NAFLD and diabetes, or cirrhosis revealed that polymeric immunoglobulin receptor (PIGR) was significantly elevated in NALFD and cirrhosis, and this result was further confirmed in mild and severe NAFLD mouse models. This study also allowed for the association of aminopeptidase N (ANPEP) and transforming growth factor-beta-induced protein ig-h3 (TGFB1) with NAFLD and cirrhosis. ANPEP and TGFB1 participate in extracellular matrix remodeling in fibrosis. These findings underly the role of plasma proteome profiling for identification of biomarkers and drug targets in liver disease [125].

Targeted proteomics allow for the simultaneous quantification of tens of proteins through the monitorization of proteotypic peptides, which are peptides that are unique and specific for a given protein (Figure 5). In selected reaction monitoring (SRM)-based targeted proteomics, proteotypic peptides are fragmented in the collision cell of the mass spectrometer, generating a specific MS2 fragmentation spectrum. The combined selection of the precursor or parental ions and several fragments or transitions, makes this technique highly specific. Additionally, the wide dynamic range of targeted proteomics, and its high sensitivity (<1 fmol), makes the method suitable for the precise quantification of the targeted protein panel in complex mixtures [126]. PRM (Parallel reaction monitoring) is also a suitable method for targeting the selected proteins by mass spectrometry. The main advantages of this method are the simplicity as the mass spectrometer is operated by providing an inclusion list and there is no need to develop a tailor-made method to monitor the selected panel of proteins. Furthermore, in contrast to MRM that requires QqQ instruments, PRM analysis is performed in high resolution analyzers that provide accurate identification of the targeted peptides based on information of all the fragment ions instead of a predetermined selection as in MRM.

Targeted proteomics (SRM) allows for the monitorization of complete cellular pathways, providing a precise and systematic measurement of their activity in any physiological or pathological condition. This approach has been applied to the study of the molecular mechanisms underlying insulin resistance in NAFLD after a sustained high-fat diet. 144 proteins involved in insulin signaling pathway, gluconeogenesis, glycolysis, tricarboxylic acids cycle, fatty acid biosynthesis, ketogenesis, and others, were specifically monitored by SRM. Interestingly, differential responses were reported for mice with different genetic backgrounds (129Sv and C57B6/J). The predominant response of C57BL/6J mice was sustained on the activation of peroxisomal beta-oxidation, while lipogenesis was the main effect of high-fat diet in 129Sv mice [127].

Using a similar approach, and in light of the central role of OCM in hepatocyte homeostasis, a method to quantify mouse OCM enzymes by targeted proteomics has been developed. The application of this method in different mice tissues revealed that the liver had high expression levels of MAT1A, GNMT, AHCY, BHMT, CBS, CGL, SHMT, MTAP, and DHFR. The expression levels of OCM enzymes in the brain and spleen was significantly lower than in adult liver, while the kidney had high levels of expression of MAT2A, MAT2B, and other OCM enzymes, likely due to the fact that this organ shares embryonic linage with the liver. Monitoring the expression levels of OCM enzymes by SRM targeted proteomics in different murine models of HCC (Aging, Mdr2 −/− and MAT1A −/−) showed that one-carbon metabolism reprogramming was different according to the etiology of the tumor. The OCM configuration suffered alterations in all three HCC models, with special differences in MAT1A −/− tissue. GNMT and BHMT were downregulated and MAT2A was upregulated in Mdr2 −/− tumors, leading to hepatocyte de-differentiation. In this model, MTAP, SHMT, and DHFR were upregulated, suggesting the activation of the methionine salvage pathway and the folate cycle to ensure SAM production. On the other hand, in MAT1A −/− mice MAT2A was moderately upregulated while MAT2B was significantly increased. MAT2B enhances MAPK/ERK pathway that promotes tumorigenesis. BHMT and GNMT were also significantly upregulated in this HCC model, suggesting an alternative mechanism to preserve methionine cycle [118].

Targeted proteomics has also been applied to the study of one-carbon metabolism in cirrhotic and HCC patients. A standardized method for the quantification of OCM enzymes in human liver has been developed [128]. The standardization was performed according to CPTAC criteria, then the lower limit of detection (LLOD) and lower limit of quantification (LLOQ), as well as the lineality response, reproducibility and repeatability of the measures were determined for each peptide [129]. Lower levels of GNMT, BHMT, SHMT1 were observed in HCC, while MAT2B and MTAP (unfavorable prognostic factors for HCC in the Human Protein Atlas), were upregulated in HCC. Interestingly, cirrhotic patients tended to present intermediate pattern of expression levels of one-carbon metabolism enzymes between control and HCC conditions. Machine learning-based analysis of those results was applied to build a predictive model for classification of samples according to their condition. Quadratic discriminants analysis achieved an in-sample accuracy of 83.05%. These results indicate that monitoring one-carbon metabolism enzymes expression levels could be useful for early diagnosis of HCC, especially in patients with cirrhosis [130].

In conclusion, one-carbon metabolism plays a key role in maintaining functional and differentiated hepatocytes, and therefore, a healthy liver. Coordinate changes of the expression of several OCM enzymes or intermediate metabolites like SAM or MTA are associated with liver injury progression and loss of hepatocyte differentiation. The pathology-specific remodeling of one-carbon metabolism makes this cycle a promising cell function-based biomarker for liver diseases like PFIC, NALFD, NASH, cirrhosis, or HCC. Targeted proteomics has proved to be a useful tool for the systematic monitorization of one-carbon metabolism enzymes in the liver without the need of antibodies.

## Figures and Tables

**Figure 1 metabolites-12-00779-f001:**
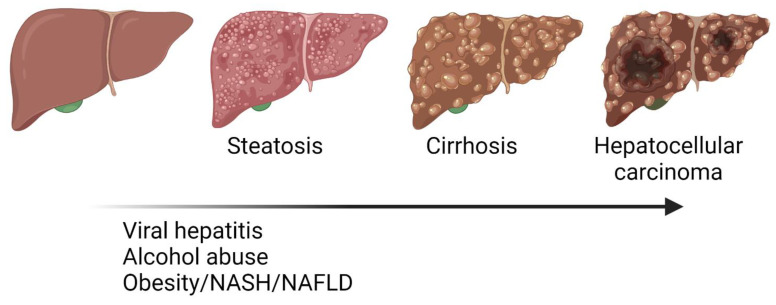
Progression from steatosis, fibrosis, cirrhosis to hepatocellular carcinoma (HCC). Liver diseases involve different pathological events including metabolic reprograming, inflammation, extracellular matrix imbalance, functional impairment, and cellular transformation. Major risk factors for liver diseases are already known but pathogenic mechanisms must still be elucidated to improve their clinical management.

**Figure 2 metabolites-12-00779-f002:**
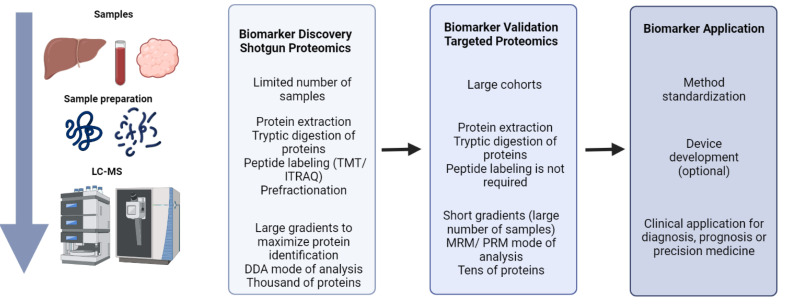
Proteomics workflow for biomarkers discovery. Sample collection, storage, protein extraction, and processing are essential steps for proteomics analysis. Shotgun analysis allows for the identification, quantification, and characterization of thousands of proteins; it is the ideal procedure for biomarker discovery. Once identified, candidates must be confirmed in large cohorts to assess their discriminatory capacity with sufficient sensitivity and specificity; this phase demands high sample throughput analytical strategies, such as targeted methods (MRM, PRM) to quantify the panel of selected proteins. Finally, the clinical application may rely on the standardization of the methods and eventually on the development of devices for a straightforward implementation in the clinical routine.

**Figure 3 metabolites-12-00779-f003:**
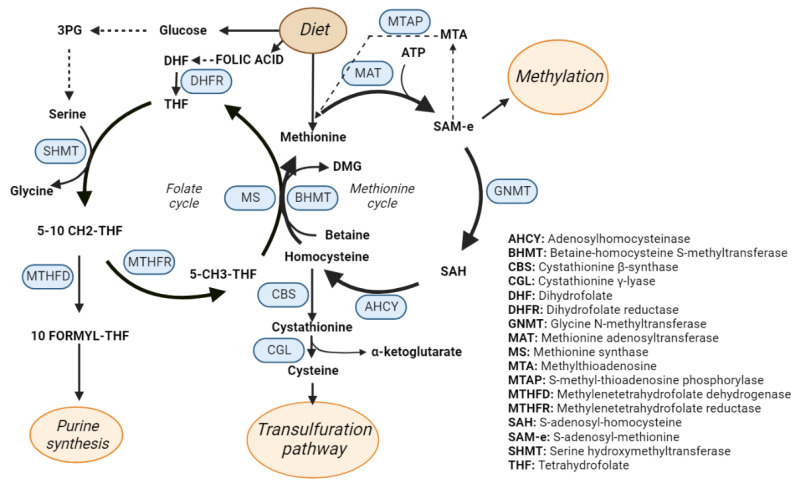
One-carbon metabolism (OCM). Schematic representation of the OCM cycle that includes the methionine and folate cycles. The enzymes catalyzing the OCM reactions are highlighted in blue.

**Figure 4 metabolites-12-00779-f004:**
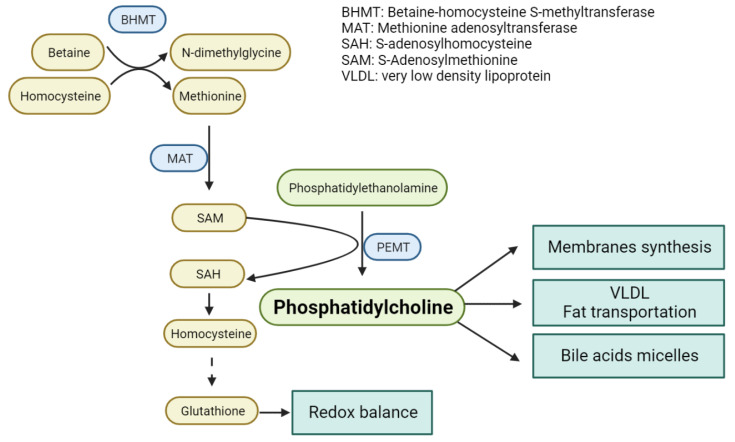
Phosphatidylcholine (PC) synthesis through one-carbon metabolism intermediates. SAM is an essential metabolite for PC synthesis as it is the methyl group donor in the three sequential methylation reactions of the precursor phosphatidylethanolamine (PE). This reaction is catalyzed by the PE N-methyltransferase.

**Figure 5 metabolites-12-00779-f005:**
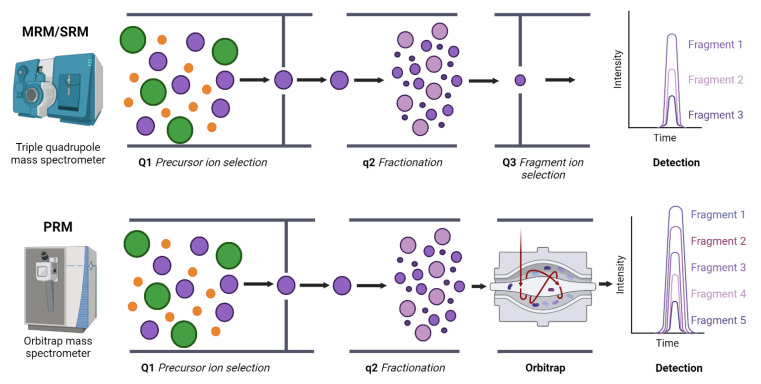
Multiple/selected reaction monitoring (MRM/SRM) and parallel reaction monitoring (PRM) targeted proteomics. MRM or SRM targeted proteomics experiments are performed in triple quadrupoles mass spectrometers while PRM experiments are performed in orbitrap high resolution mass spectrometers. In PRM experiments, intensity values of all fragments from a precursor ion are available, allowing a more robust quantification.

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
