# Peer review of "Targeted Proteomics for Monitoring One-Carbon Metabolism in Liver Diseases"

_metabolites, 2022, doi:10.3390/metabo12090779_

Round 1

Reviewer 1 Report

This review summarized the liver diseases, one-carbon metabolism, and how proteomics was used to study the biomarks linking with one-carbon metabolism and liver diseases. The review is help for readers understanding the the progress of liver diseases and the regulation of one-carbon metabolism.

From the title of this manuscript, authors want to review that targeted proteomics was used to monitoring the changes of proteins and enzymes which take part in one-carbon metabolism. While there are too many words of summarizing the liver diseases. This parts should be shorted and improved and the discusstion of targeted proteomics should be strengthened.

From line 156 to 172, authors reviewed the msass spectrometry techniques can be applied for biomarker discovery and validation. Besides the Data-dependent Acquisition(DDA), the Data-independent Acquisition(DIA) is important method to discover the biomarks of different diseases, expecially the cohort studies. It should be discussed in this review.

For the targeted proteomics, the diversity of mass spectrometers for targeted proteins analysis(QQQ, IT, TOF, Orbitrap and others), the advantages and disadvantages of MRM and PRM, the separation of peptides before MS acquisition(Nano-, Capillary- and Micro- LC) , and the ion mobility(FAIMS and timsMS) should also be discussed in modern and further targeted quantitative applications.

Author Response

Reviewer 1

This review summarized the liver diseases, one-carbon metabolism, and how proteomics was used to study the biomarks linking with one-carbon metabolism and liver diseases. The review is help for readers understanding the the progress of liver diseases and the regulation of one-carbon metabolism.

From the title of this manuscript, authors want to review that targeted proteomics was used to monitoring the changes of proteins and enzymes which take part in one-carbon metabolism. While there are too many words of summarizing the liver diseases. This parts should be shorted and improved and the discusstion of targeted proteomics should be strengthened.

We thank the reviewer comments. Modifications in the text have been done accordingly

From line 156 to 172, authors reviewed the msass spectrometry techniques can be applied for biomarker discovery and validation. Besides the Data-dependent Acquisition (DDA), the Data-independent Acquisition(DIA) is important method to discover the biomarks of different diseases, expecially the cohort studies. It should be discussed in this review.

The review was intended to discuss the application of proteomics to the understanding of liver disorders, focusing in the role OCM. Proteomics technologies were not described in depth to avoid dilution of the main biological topic. However, as the reviewer indicated, DIA is a relevant method in biomarker discovery and therefore it has been included in the new version of the manuscript.

For the targeted proteomics, the diversity of mass spectrometers for targeted proteins analysis(QQQ, IT, TOF, Orbitrap and others), the advantages and disadvantages of MRM and PRM, the separation of peptides before MS acquisition(Nano-, Capillary- and Micro- LC) , and the ion mobility(FAIMS and timsMS) should also be discussed in modern and further targeted quantitative applications. 

As mentioned in the previous point, we prioritized the discussion on how technology has been applied to the study of liver diseases over detailed technical discussions. However, following the reviewer recommendation, we have introduced a few considerations about the main technical alternatives that integrate the proteomics toolbox.

Reviewer 2 Report

In the review „reprogramming of one-carbon metabolism in liver diseases is described and the role of mass spectrometry to follow-up these alterations is discussed”.

The topics of the paper are approached in a new and original light, i.e. in relation to the liver and primary liver cancer, namely hepatocellular carcinoma (HCC). The importance of the search for still more ideal markers for the early diagnosis of HCC is emphasized, as this cancer still leaves too low a percentage of patients with 5-year survival. The review paper is in a descriptive form, the subsections are logical and written in an understandable way for the reader. The works cited are recent, hence this aspect is also important to emphasize.

Minor comments:

To improve the quality and professionalism of the work, minor comments:

1. please correct (add, move or delete) some abbreviations, e.g. in lines 71-72, line 394 (add), line 221 (move the abbreviation description from line 241 here), line 305 (delete because it was already in line 301), line 309 (delete because it was already in line 221), the abbreviation "CBS" in line 327 (delete because it was already on page 7 in line 229); you can replace the whole name in lines 324-325 with the abbreviation CBS; 

2. I also recommend expanding the legends to all figures (but especially Fig. 3 and Fig. 4), by including an explanation of the abbreviations present in the figures and/or a brief description of the signaling pathways placed, so that the figure is "self-sufficient" for understanding the processes/stages, etc. depicted in it. Please include whole names in figure headings (e.g., Fig. 1, HCC). Figure 4 please correct by highlighting "phosphatidylcholine" (larger frame, different color??), there is also in this figure a minor error in the word" Fats transportation";

3. standardize the abbreviation "OCM" and "1CM" throughout the text, as it is sometimes confusing;

4. suggest organizing the abbreviations on page 15 in alphabetical order;

5. please also standardize the spelling of proteins and genes (the latter in lower case and in italics);

6. other minor spelling errors noted: line 143-"HVC" - and should be "HCV"; line 343 should be "demonstrating that the regulation"; line 419-remove one word "of"; line 437-remove the period after the word "and"; line 482- add a period after the word "production".

In summary: The review has a practical aspect and promises citations. The corrections need especially more professional description of figures.

Author Response

Reviewer 2

In the review „reprogramming of one-carbon metabolism in liver diseases is described and the role of mass spectrometry to follow-up these alterations is discussed”. 

The topics of the paper are approached in a new and original light, i.e. in relation to the liver and primary liver cancer, namely hepatocellular carcinoma (HCC). The importance of the search for still more ideal markers for the early diagnosis of HCC is emphasized, as this cancer still leaves too low a percentage of patients with 5-year survival. The review paper is in a descriptive form, the subsections are logical and written in an understandable way for the reader. The works cited are recent, hence this aspect is also important to emphasize. 

Minor comments:

To improve the quality and professionalism of the work, minor comments: 

  1. please correct (add, move or delete) some abbreviations, e.g. in lines 71-72, line 394 (add), line 221 (move the abbreviation description from line 241 here), line 305 (delete because it was already in line 301), line 309 (delete because it was already in line 221), the abbreviation "CBS" in line 327 (delete because it was already on page 7 in line 229); you can replace the whole name in lines 324-325 with the abbreviation CBS;  

We thank the reviewer for these comments. Changes have been now made in the revised manuscript

  1. I also recommend expanding the legends to all figures (but especially Fig. 3 and Fig. 4), by including an explanation of the abbreviations present in the figures and/or a brief description of the signaling pathways placed, so that the figure is "self-sufficient" for understanding the processes/stages, etc. depicted in it. Please include whole names in figure headings (e.g., Fig. 1, HCC). Figure 4 please correct by highlighting "phosphatidylcholine" (larger frame, different color??), there is also in this figure a minor error in the word" Fats transportation";

Changes have been done as suggested. Thank you.

  1. standardize the abbreviation "OCM" and "1CM" throughout the text, as it is sometimes confusing;

Done as indicated, Thank you.

  1. suggest organizing the abbreviations on page 15 in alphabetical order;

Done as indicated, Thank you.

  1. please also standardize the spelling of proteins and genes (the latter in lower case and in italics);

Done as indicated, Thank you.

  1. other minor spelling errors noted: line 143-"HVC" - and should be "HCV"; line 343 should be "demonstrating that the regulation"; line 419-remove one word "of"; line 437-remove the period after the word "and"; line 482- add a period after the word "production". 

Done as indicated, Thank you.

In summary: The review has a practical aspect and promises citations. The corrections need especially more professional description of figures.

Round 2

Reviewer 1 Report

The authors have significantly improved the manuscript.  I think that the presented manuscript worth publication.